# Artificial HfO_2_/TiO_x_ Synapses with Controllable Memory Window and High Uniformity for Brain-Inspired Computing

**DOI:** 10.3390/nano13030605

**Published:** 2023-02-02

**Authors:** Yang Yang, Xu Zhu, Zhongyuan Ma, Hongsheng Hu, Tong Chen, Wei Li, Jun Xu, Ling Xu, Kunji Chen

**Affiliations:** 1School of Electronic Science and Engineering, Nanjing University, Nanjing 210093, China; 2Collaborative Innovation Center of Advanced Microstructures, Nanjing University, Nanjing 210093, China; 3Jiangsu Provincial Key Laboratory of Photonic and Electronic Materials Sciences and Technology, Nanjing University, Nanjing 210093, China

**Keywords:** artificial synapse, memristor, brain-inspired computing

## Abstract

Artificial neural networks, as a game-changer to break up the bottleneck of classical von Neumann architectures, have attracted great interest recently. As a unit of artificial neural networks, memristive devices play a key role due to their similarity to biological synapses in structure, dynamics, and electrical behaviors. To achieve highly accurate neuromorphic computing, memristive devices with a controllable memory window and high uniformity are vitally important. Here, we first report that the controllable memory window of an HfO_2_/TiO_x_ memristive device can be obtained by tuning the thickness ratio of the sublayer. It was found the memory window increased with decreases in the thickness ratio of HfO_2_ and TiO_x_. Notably, the coefficients of variation of the high-resistance state and the low-resistance state of the nanocrystalline HfO_2_/TiO_x_ memristor were reduced by 74% and 86% compared with the as-deposited HfO_2_/TiO_x_ memristor. The position of the conductive pathway could be localized by the nanocrystalline HfO_2_ and TiO_2_ dot, leading to a substantial improvement in the switching uniformity. The nanocrystalline HfO_2_/TiO_x_ memristive device showed stable, controllable biological functions, including long-term potentiation, long-term depression, and spike-time-dependent plasticity, as well as the visual learning capability, displaying the great potential application for neuromorphic computing in brain-inspired intelligent systems.

## 1. Introduction

With the big data era coming, artificial neural networks, as a game-changer to break up the bottleneck of classical von Neumann architectures, have attracted great interest due to their high massive information processing speed [1,2,3]. As a unit of artificial neural networks, memristive devices play a key role in achieving a biological function because of their similarity to biological synapses in structure, dynamics, and electrical behaviors [4,5,6,7,8,9]. To ensure that artificial synapses can be used in neuromorphic computing, memristive devices with a tunable memory window and high uniformity are vitally important. Among the candidates for artificial synapses, HfO_2_/TiO_x_ memristors are preferred for their compatibility with CMOS and remarkable characteristics of resistance switching [10,11,12,13,14,15,16,17,18,19,20,21]. Compared with memristors based on the low-dimensional structure of TiO_x_ from nanoparticles to nanorods [22,23,24], the advantage of HfO_y_/TiO_x_ bilayers lies in two main aspects. First, the barrier of the HfO_2_ and TiO_x_ sublayer can be modulated by tuning the dielectric permittivity of TiO_x_ (~80) and HfO_2_ (~22) [25]. Second, the conductive pathway’s morphology can be controlled by tuning the thicknesses of the HfO_2_ and TiO_x_ sublayers due to the various concentrations of oxygen vacancies in the two kinds of sublayers. To obtain an HfO_2_/TiO_x_ bilayer memristor with a tunable memory window and high uniformity, we fabricated HfO_2_/TiO_x_ bilayers with different thickness ratios by the atomic layer deposition (ALD) method. Compared with the as-deposited HfO_2_/TiO_x_ bilayer, the coefficients of variation in the high-resistance state and the low-resistance state of nanocrystalline HfO_2_/TiO_x_ memristors could be reduced by 74% and 86%. It was found that the improved uniformity of the HfO_2_/TiO_x_ memristive device was related to the permanent nanocrystalline HfO_2_ and TiO_x_ dots, which could localize the position of the conductive pathway in the HfO_2_/TiO_x_ bilayer. The disconnection and formation of conductive pathways occurred mainly in the thin TiO_x_ layer, which led to a substantial improvement in the switching uniformity. When the thickness ratio of the HfO_2_/TiO_x_ sublayer increased from 1:4 to 4:1, the memory window increased from 1.0 to 1.8 V. The nanocrystalline HfO_2_/TiO_x_ bilayer memristor also exhibited multiple conductance states similar to the weight of biological synapses updating under pulses with different voltage widths and numbers. The stable, controllable operation and multilevel characteristics enabled the successful implementation of biological functions, including long-term potentiation, long-term depression, and spike-time-dependent plasticity, as well as visual capability, demonstrating the great potential application for neuromorphic systems in the AI period.

## 2. Experimental Details

Ahead of preparing the thin films, the Si substrate was cleaned following a standard Radio Corporation of American cleaning method. Then, the Ti/Pt bilayers were deposited on the P–Si substrate via electron beam evaporation as the bottom electrode. The thicknesses of the Ti and Pt were 5 nm and 40 nm, respectively. Subsequently, the HfO_2_/TiO_x_ bilayers were grown on the surface of the Ti/Pt bilayer in an atomic layer deposition (ALD) system with Hf[N(C_2_H_5_)CH_3_]_4_(TEMAH) and Ti[N(CH_3_)_2_]_4_ (TDMAT) as the main precursors. The different thickness ratios of TiO_x_ and HfO_2_ were achieved by tuning the deposition time. Three kinds of HfO_2_/TiO_x_ bilayers with different thickness ratios, including 4:1, 1:1, and 1:4, were obtained, which were named H12T3, H7.5T7.5, and H3T12. The thicknesses of HfO_x_ in H12T3, H7.5T7.5, and H3T12 were 12, 7.5, and 3 nm. The thicknesses of TiO_2_ in H12T3, H7.5T7.5, and H3T12 were 3 nm, 7.5 nm, and 12 nm. The total thickness of the HfO_2_/TiO_x_ bilayers remained at 15 nm. To obtain nanocrystalline HfO_2_/TiO_x_, the H12T3 was annealed at 700 °C for 30 min under a pure N_2_ atmosphere. After thermal annealing, the 40 nm-thick Ti top electrodes were deposited by using electron beam evaporation through a shadow mask. The atomic concentration ratios of the HfO_2_/TiO_x_ bilayer were determined through an XPS test using a PHI 5000 Versa Probe (Ulvac-Phi Inc., Chigasaki, Japan). The microstructure of the annealed H12T3 was analyzed by high-resolution cross-section transmission electron microscopy (HRXTEM) with a JEOL 2100F electron microscope (JEOL Inc., Tokyo, Japan) operated at 200 kV. An Agilent B1500A semiconductor (Agilent Inc., Santa Clara, CA, USA) analyzer was used to explore the electrical behaviors of the HfO_2_/TiO_x_ memristor under the atmosphere.

## 3. Results and Discussion

Figure 1a shows a schematic diagram and the measurement setup of the HfO_2_/TiO_x_-bilayer memristor device. An HRXTEM micrograph (JEOL Inc., Tokyo, Japan) of an annealed H12T3 bilayer is displayed in Figure 1b. The HfO_2_/TiO_x_ bilayer with a clear interface can be observed between the Ti bottom electrode (BE) and the Pt top electrode (TE). The thicknesses of HfO_2_ and TiO_x_ sublayers were 12 nm and 3 nm, respectively. It is obvious that nanocrystalline HfO_2_ and TiO_x_ dots were formed in both the HfO_2_ and TiO_x_ sublayers. The crystalline lattice parameters of the nanocrystalline TiO_x_ and HfO_2_ were 0.35 nm and 0.32 nm, respectively. It is interesting that the nanocrystalline TiO_x_ and HfO_2_ dots were connected at the interface of TiO_x_ and HfO_2_, which supplied a channel for resistive switching.

The atomic configuration of the annealed H12T3 bilayers was analyzed through the XPS spectra in Figure 1. The XPS spectrum of the HfO_2_ sublayer can be fit by two binding energy peaks located at 16.5 eV and 18.1 eV, as shown in Figure 1c, which correspond to Hf^4+^ 4f_7/2_ and Hf^4+^ 4f_5/2_, respectively. There was only one peak located at 530.2 eV, as presented in Figure 1d, which originated from O 1s. The existence of Hf^4+^ and O^2−^ in the XPS spectra indicates that the stoichiometry of the HfO_2_ sublayer met the standard chemical ratio. Regarding the TiO_x_ sublayer, the corresponding XPS spectra are displayed in Figure 1e,f. Two peaks at 458.7 eV and 464.5 eV were detected, which were related to Ti^4+^ 2p_3/2_ and Ti^3+^ 2p_3/2_, respectively [26,27,28,29,30]. The atomic percentages of Ti^4+^ and Ti^3+^ were 75.2% and 24.8%. As shown in Figure 1f, the peak corresponding to O 1s could be fitted with two peaks, including lattice oxygen (at 530.1 eV) and non-lattice oxygen (at 531.7) [31,32]. The atomic percentages of lattice oxygen and non-lattice oxygen were 74.5% and 25.5%. They correspond to the oxide component and defective oxides, respectively. The defect-induced oxidation state of Ti^3+^ verified the existence of oxygen vacancies. It is worth noting that the atomic percentage of Ti^3+^ in the Ti 2p spectra was 24.8% and the atomic percentage of the oxygen vacancies in the O 1s spectra was 25.5%. The high concentrations of Ti^3+^ and oxygen vacancies reveal that a large number of oxygen vacancies coexisted with the nanocrystalline TiO_x_ and HfO_2_ dots in the TiO_x_ sublayer.

Figure 2a displays the electroforming characteristics of the as-deposited HfO_2_/TiO_x_ memristor with different thickness ratios of HfO_2_ and TiO_x_ during the positive DC sweeping with a compliance current (Icc) of 10 mA. It can be observed that the forming voltage increased from 2.3 V to 4.4 V as the thickness of TiO_x_ decreased from 12 nm to 3 nm. This was because the number of oxygen vacancies in the TiO_x_ sublayer was higher than that of the HfO_2_ sublayer, which made the main contribution to the formation of the conductive pathway. As the thickness of the TiO_x_ sublayer decreased, the diameter of the oxygen conductive pathway became thinner. Thus, the total resistance of the HfO_2_/TiO_x_ memristor increased, which is the reason why a higher voltage was needed to complete the forming process. The typical bipolar resistive switching characteristic was detected from the three devices with different HfO_2_/TiO_x_ thickness ratios, as shown in Figure 2b. The set process was observed under a positive bias voltage, while the reset process occurred under a negative bias voltage. The resistances of the low-resistance state (LRS) for all three devices were basically consistent, while the resistances of the high-resistance state (HRS) increased gradually as the thickness of the HfO_2_ sublayer increased from 3 to 12 nm. The cell-to-cell distribution of the memory window for the three kinds of devices is shown in Figure 2c. It is obvious that the memory window increased to 100 as the thickness of TiO_x_ decreased from 12 nm to 3 nm. The H12T3 device showed the largest memory window compared with the other two devices, which was related to the largest resistance of the HRS.

Considering that the H12T3 device had the largest memory window, it was chosen to be annealed to form the nanocrystalline HfO_2_/TiO_x_ device. The resistive switching characteristics of the nanocrystalline HfO_2_/TiO_x_ device are displayed in Figure 2d. Good uniformity and repeatability were observed for the nanocrystalline HfO_2_/TiO_x_ device after 100 switching cycles compared with those of the as-deposited device. The statistical distribution of the set and reset voltages is displayed in Figure 2e,f. It is worth noting that the average set voltages were reduced from 1.53 V to 1.02 V after annealing. Meanwhile, the average reset voltage decreased from −1.46 V to −0.8 V after annealing. The distribution consistency of the set voltage decreased from 0.25 to 0.08 after annealing. To analyze the variability in the set voltage and reset voltage after 100 cycles, we calculated the coefficient of variation, which was equal to the ratio of the standard deviation (σ) and the mean value (u) of V_SET_ or V_RESET_ in absolute value. Compared with the as-deposited HfO_2_/TiO_x_ bilayer, the coefficient of variation in the high-resistance state and the low-resistance state of the nanocrystalline HfO_2_/TiO_x_ memristor were reduced 74% and 86%. The endurance and retention characteristics of the nanocrystalline HfO_2_/TiO_x_ device are presented in Figure 2g,h. It is evident that the resistances of the HRS and LRS for the nanocrystalline HfO_2_/TiO_x_ device were retained well after 100 cycles, exhibiting good stability compared with the as-deposited HfO_2_/TiO_x_ device. The memory window of 100 could be retained after 10^4^ s, indicating good retention characteristics. As the set voltage increased from −1.5 to −2.0 V, multiple resistance states were obtained, as shown in Figure 2i. The tunable resistance state is beneficial for constructing artificial synapses for neuromorphic computing.

To investigate the resistive switching mechanism of the HfO_2_/TiO_x_ RRAM device, the resistive switching characteristics of the H12T3 device before and after annealing are plotted in Figure 3a,b. The set process characteristics of the H12T3 device before and after annealing are replotted in double logarithmic scales, as shown in Figure 3c,d. In the low-voltage area of the HRS, the slopes of the H12T3 device before and after annealing were 1.09 and 1.15, respectively. The carrier transportation was in agreement with the ohmic conduction model. The thermal excitation in the conduction band was the origin of the mobile electrons. As the voltage increased, the initial region of the HRS showed slopes of 1.78 and 1.84, corresponding to the devices before and after annealing, respectively. The carrier transportation obeyed the SCLC model. After switching to LRS, the slope of the H12T3 device before and after annealing decreased from 1.78 and 1.84 to 1.15 and 1.04, respectively. This means that the ohmic conduction mode plays a dominant role in carrier transportation.

To reveal the relationship between the TiO_x_ thickness and the diameter of the conductive filament in more detail, we produced a schematic resistive pathway model related to different thickness ratios of TiO_x_/HfO_x_ in an RRAM device. As shown in Figure 4a, there were many oxygen vacancies in the TiO_x_ sublayer of the TiO_x_/HfO_x_ memristor with a thickness ratio of 4/1 in its initial state. When positive bias was applied to the Ti top electrode, more oxygen vacancies could be produced as the oxygen ions moved toward the Pt bottom electrode. Thus, a thicker conductive pathway could be formed in the TiO_x_ sublayer, which combined with the oxygen vacancies in the HfO_x_ sublayer to form a gradual conductive pathway. In contrast, the number of oxygen vacancies in the thinner TiO_x_ sublayer decreased as the thickness ratio of the TiO_x_ and HfO_x_ decreased from 4/1 to 1/4. Therefore, a thinner conductive pathway could be constructed in the TiO_x_ sublayer, which connected to the oxygen vacancies in the HfO_x_ sublayer to form the thinnest conductive pathway, as shown in Figure 4b. When negative bias was applied on the Ti top electrode, the oxygen ions near the bottom electrode could migrate toward the top electrode to neutralize the oxygen vacancies, causing the conductive pathway to disconnect. Therefore, the TiO_x_/HfO_x_ RRAM device could be switched from the LRS to the HRS. The gradual change in conductance is similar to the synaptic weight update in a biological synaptic device.

Figure 5a,b show the resistive switching pathway model of the RS behaviors in the H12T3 device before and after annealing. For the as-grown HfO_2_/TiO_x_ memristor, oxygen vacancies existed in the TiO_x_ sublayer at the initial state. Under a positive voltage, the electric field could drive the O^2−^ to move to the Ti electrode and leave a great number of oxygen vacancies in the top electrode; thus, Ti is an oxygen affinity electrode. Once the oxygen vacancies in the thin film reached a high level, the thin, tapered conductive filament composed of oxygen vacancies in the HfO_2_ film could connect to the thick conductive filament in the TiO_x_ film, switching the device to the LRS. Under the negative bias, O^2−^ could move back to the Pt electrode from the boundary of the Ti electrode and neutralize the oxygen vacancies, which would break up the oxygen vacancy conductive filament. As the distribution of oxygen vacancies was uneven in the TiO_x_ sublayer, the position of the conduction pathway varied as the cycles increased. In contrast to the TiO_x_/HfO_x_ device before annealing, nanocrystalline TiO_x_ and HfO_x_ were formed in the TiO_x_ and HfO_x_ sublayers after annealing, as revealed by the HRTEM in Figure 1. Nanocrystalline TiO_x_ and HfO_x_ acted as a presetting conductive channel, which could be connected to oxygen vacancies to form the whole conductive pathway under positive bias. The introduction of nanocrystalline TiO_x_ and HfO_2_ supplied a “fixed position” to confine the growth and rupture of oxygen vacancies during the RS behaviors. In contrast to the device after annealing, the distribution of the oxygen vacancy conductive filament under a positive bias was random in the as-deposited device without the confinement provided by nanocrystalline TiO_x_ and HfO_2_, as shown in Figure 3c. Under a negative bias, not all oxygen vacancies in the nanopathway were accurately neutralized by O^2−^, which resulted in fluctuations in the switching parameters during the successive cycles. It is evident that nanocrystalline TiO_x_ and HfO_2_ are beneficial for improving the uniformity of conductive nanopathways, while the formation and neutralization of oxygen vacancies were responsible for the bridging and rupture of conductive nanopathways. We also noticed that the set voltage of the device after annealing was reduced as the conductivity of nanocrystalline TiO_x_ and HfO_2_ was higher than that of amorphous TiO_x_ and HfO_2_. It was further revealed that nanocrystalline TiO_x_ and HfO_2_ were the main contributors to the conductive pathway.

A schematic illustration of an artificial synapse based on the nc-HfO_2_/TiO_x_ device is shown in Figure 6a. The changeable conductance of the nc-HfO_2_/TiO_x_ memristor was analogous to the connection strength between biological synapses [33,34]. The long-term potentiation (LTP) and long-term depression (LTD) characteristics of nc-HfO_2_/TiO_x_ were detected by adjusting the pulse amplitude and numbers, as shown in Figure 6b,c. Regarding the potentiation process, the conductance continued to increase as the pulse amplitude increased from 0.7 to 1.25 V under the same pulse width of 100 us. This indicates that the strength of the synaptic connection between two neurons was strengthened. It is noteworthy that the conductance tended to be saturated when the pulse number increased to 20. Regarding the depression process, the conductance continued to decrease as the negative pulse amplitude increased from −1 to −1.32 V under the same pulse width of 100 us. This meant that the strength of the synaptic connection between two neurons was weakened. The rule of the depression process was consistent with the potentiation process in general. Figure 6d shows the spike-timing-dependent plasticity (STDP) of nc-HfO_2_/TiO_x_ synapses, which is an important rule for learning and memory. The synaptic weight could be modulated by the relative timing of the pre-spike and the post-spike, which was determined by the gradual change in the conductance. As revealed in Figure 6d, when the pre-spike was ahead of/dropped behind the post-spike, the connection strength of the synapse between the two neurons was strengthened/weakened. It can be seen that a larger conductance change was caused by a closer spike timing, which was consistent with the Hebbian learning rule. The total conductance change in nc-HfO_2_/TiO_x_ synapses is defined as ΔG, which can be expressed as the following equation [35]:

(|G_after-G_before |)/(Min(G_after,G_before))
(1)$$\Delta G=\frac{\left|G_{\mathrm{after}}-G_{\mathrm{before}}\right|}{\mathrm{Min}\left(G_{\mathrm{after}},G_{\mathrm{before}}\right)}$$
where G_before_ and G_after_ are the conductance before and after the application of the spike. ΔG exhibited an increasing tendency when the pre-spike was ahead of the post-spike (Δt > 0) during the potentiation process, while it displayed a decreasing tendency when the post-spike was ahead of the pre-spike (Δt < 0) during the depression process. The pulse response of nc-HfO_2_/TiO_x_ synapses showed that the maximum value of ΔG could reach 650% for potentiation and 1400% for depression. The successful implementation of biosynaptic function ensured further application in neuromorphic computing. In order to visualize the information transferred to the HfO_2_/TiO_x_ neural network, 5 × 5 synaptic arrays based on the Ti/TiO_x_/HfO_2_/Pt memristor were applied to image recognition, as shown in Figure 6e. The conductance is represented by the color level. In the initial state, the conductance of all synapses ranged randomly. A unit device distributed in the shape of a “T” in the synapse array was selected to input 10 consecutive stimulus pulses with a duration of 1 us and amplitude of 1.15 V. The conductance of the HfO_2_/TiO_x_ unit device increased obviously after the application of stimulus pulses. Then, the number of consecutive stimulus pulses increased from 10 to 15, 20, and 25. After training by the 25 consecutive stimulus pulses with the same duration and amplitude, the image clearness of “T” reached a high level, as shown in the lower panel of Figure 6e. The visualized evolution of weight values showed that the HfO_2_/TiO_x_ memristor crossbar arrays had the potential to mimic real neurotransmission in a biological system.

**Figure 6 nanomaterials-13-00605-f006:**
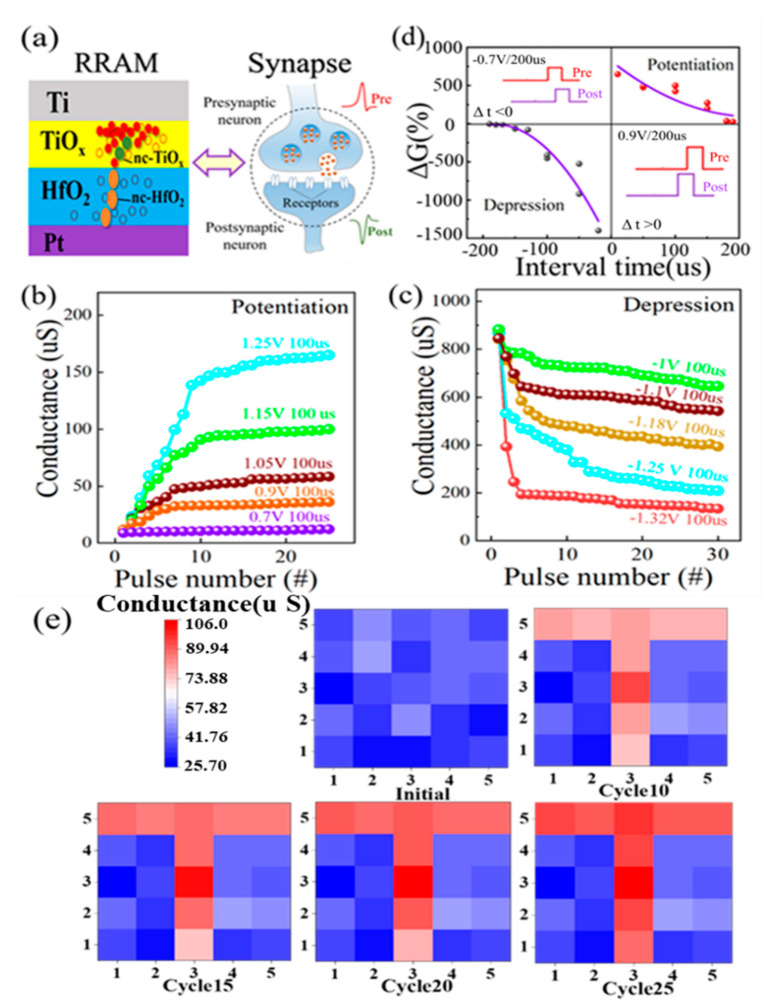
(**a**) Schematic illustration of the similarity between a biological synapse and an HfO_2_/TiO_x_ memristor, displaying the diffusion of neurotransmitters among synaptic neurons and an electrical synapse based on the top electrode, switching matrix, and bottom electrode. (**b**,**c**) Long-term potentiation and depression of HfO_2_/TiO_x_ synapse by changing the pulse amplitude under the same pulse duration; (**d**) spike-timing-dependent plasticity of HfO_2_/TiO_x_ synapse as a function of interval time between pre-spike and post-spike; (**e**) simulated image memorization of HfO_2_/TiO_x_ synapse under consecutive pulses on the selected synapse.

## 4. Conclusions

In summary, we successfully obtained an artificial synapse based on an HfO_2_/TiO_x_ memristive crossbar with a controllable memory window and high uniformity. The controllable memory window could be obtained by tuning the thickness ratio of the sublayers. The position of the conductive pathway could be localized by the nanocrystalline HfO_2_ and TiO_2_ dot, leading to a substantial improvement in the switching uniformity. Notably, the coefficient of variation in the high-resistance state and the low-resistance state of the nanocrystalline HfO_2_/TiO_x_ memristor could be reduced by 74% and 86% compared with those of the as-deposited HfO_2_/TiO_x_ memristor. The nanocrystalline HfO_2_/TiO_x_ memristive device showed stable, controllable biological functions, including long-term potentiation, long-term depression, and spike-time-dependent plasticity, as well as visual learning capability, which provides a hardware base for their integration into the next generation of brain-inspired chips.

## Figures and Tables

**Figure 1 nanomaterials-13-00605-f001:**
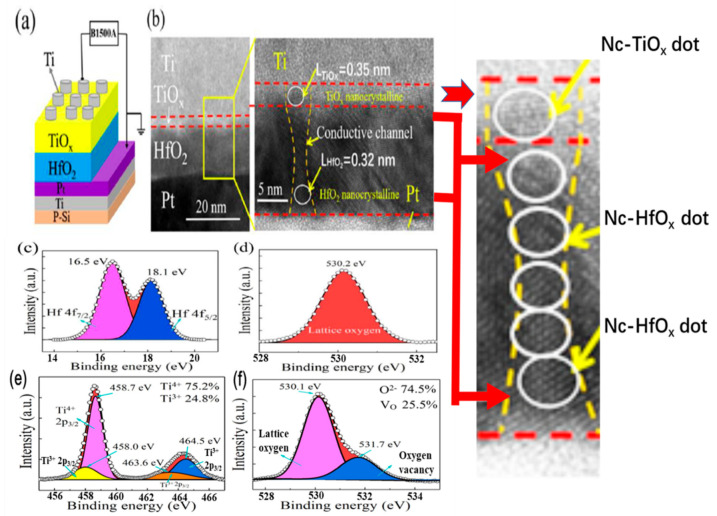
(**a**) Schematic diagram of Ti/TiO_x_/HfO_2_/Pt memristor device; (**b**) cross-sectional TEM image of H12T3 bilayers after annealing; (**c**,**d**) XPS spectra of Hf 4f and O1s in the HfO_2_ sublayer; (**e**,**f**) XPS spectra of O1s andTi 2p in the TiO_x_ sublayer.

**Figure 2 nanomaterials-13-00605-f002:**
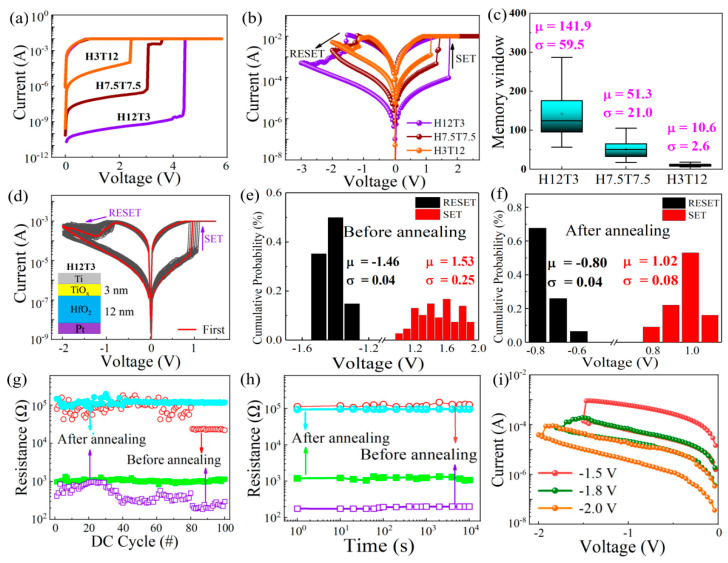
(**a**,**b**) Formation characteristics and bipolar resistive switching of the HfO_2_/TiO_x_ memristor with different sublayer thickness ratios from 4/1 to 1/4, including the devices named H12T3, H7.5T7.5, and H3T12; (**c**) memory window distribution of the HfO_2_/TiO_x_ memristors with different sublayer thickness ratios from 4/1 to 1/4, including the devices named H12T3, H7.5T7.5, and H3T12; (**d**) I-V characteristics of the H12T3 memristor after annealing; (**e**,**f**) statistical distribution of the set/reset voltage for the H12T3 memristor before and after annealing; (**g**,**h**) endurance and retention characteristics of the H12T3 memristor before and after annealing; (**i**) multilevel I–V characteristics of the RRAM device under different reset voltages.

**Figure 3 nanomaterials-13-00605-f003:**
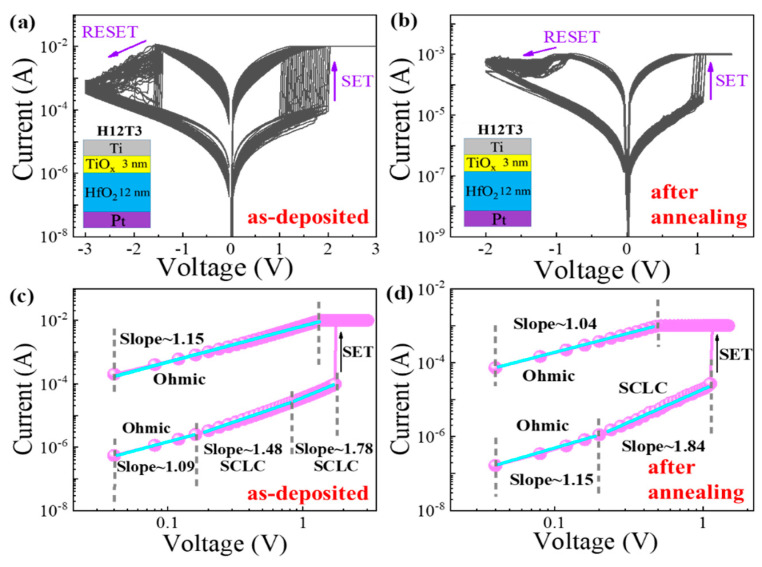
(**a**,**b**) Resistive switching characteristics of HfO_2_/TiO_x_ RRAM (H12T3) before and after annealing; (**c**,**d**) set process characteristics of the H12T3 device before and after annealing replotted in double logarithmic scales.

**Figure 4 nanomaterials-13-00605-f004:**
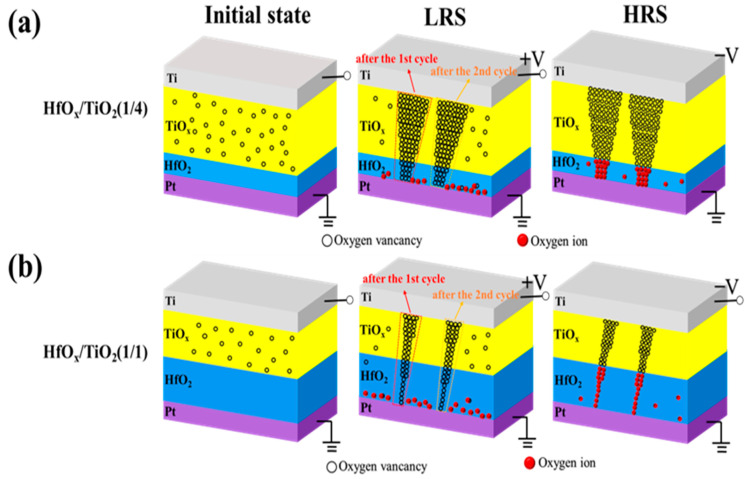
(**a**,**b**) Schematic resistive switching model of as-grown HfO_2_/TiO_x_ memristor with thickness ratio of 1/4 and 1/1 corresponding to HfO_x_ and TiO_x_ sublayer, respectively.

**Figure 5 nanomaterials-13-00605-f005:**
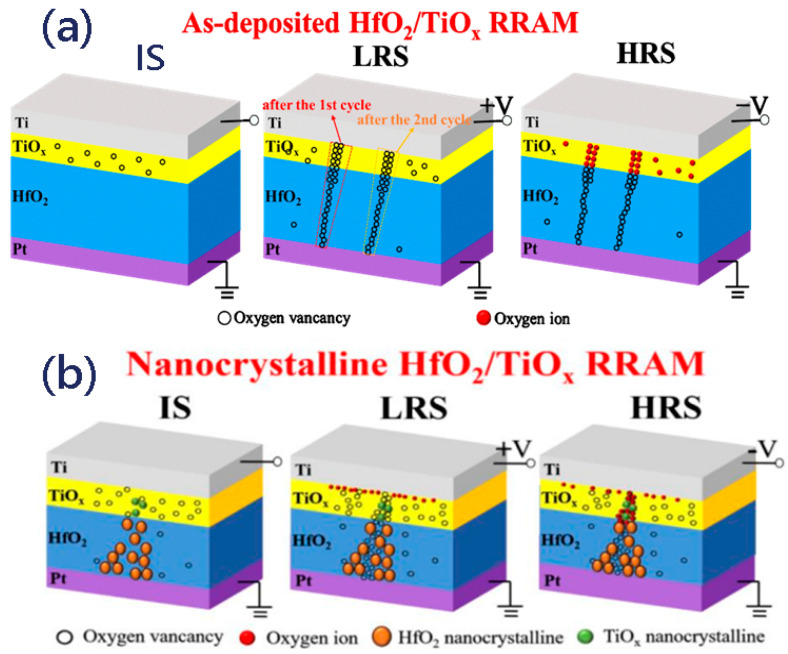
(**a**,**b**) Schematic description of the resistive switching pathway model of H12T3 before and after annealing.

## Data Availability

The data that support the findings of this study are available from the corresponding authors upon reasonable request.

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
