# Peer review of "Artificial HfO2/TiOx Synapses with Controllable Memory Window and High Uniformity for Brain-Inspired Computing"

_nanomaterials, 2023, doi:10.3390/nano13030605_

Round 1

Reviewer 1 Report

The authors present results on fabrication/characterization and electronic properties of thin film bi-layer memristive devices for different layer thickness ratios in this manuscript.  Overall, the manuscript contains some potentially interesting and useful results but is not clear in certain parts.  The following points need to be addressed before a decision can be made:

1. It is claimed with reference to the microscope images of fig.1 in the manuscript that dots are connected, which provides the conductive channel for switching.  This is difficult to follow and does not appear obvious from the data presented.  The authors need to clarify/show how they determine where the dots are and further, why they claim there is a conductive channel connecting them.  At the moment it is not clear.

2. It is unclear why the authors fixed the total thickness to 15 nm.  Is this critical to the behavior observed?  In addition, most of the detailed data focuses on the 4:1 device only; why weren't other ratios presented as well (e.g., 2:1 or 10:1, etc.)?

3. As one of the main results of this work is the tunable memory behavior for different thickness ratios (shown in figure 2), it would be much more useful and relevant from a research point of view if the schematic resistive pathway model shown in figs.3c, d compared different thickness ratio devices.

4. Please clearly state how the coefficient of variation percentages stated in the manuscript were obtained.  They are unclear and/or inconsistent.

Minor points - i. The sample labels used in the manuscript are somewhat confusing to the reader and it would be helpful if more descriptive labels were used.  ii. Please provide the definition of memory window used in the manuscript.

Reviewer 2 Report

In this paper, the authors reported the synaptic memristor device with the structure of nanocrystalline HfO2/TiOx. With the optimized thickness ratio of HfO2 and TiOx, the memristive device shows a large memory window and uniform resistive switching. Also, the device successfully emulates the biological synaptic functions such as potentiation-depression characteristics and spike-timing-dependent plasticity (STDP) essential for learning the artificial neural networks. However, this reviewer does not fully agree with the reason for improving the uniformity switching that the authors suggest and further clarification on their experimental results is needed. Therefore, this paper is not ready yet for publication. Detailed comments are:

 1. The authors mentioned that decreasing the thickness of TiOx reduces the diameter of oxygen vacancy-based conductive filament. However, considering that the resistance for HfO2 is larger than the resistance of TiOx, increasing the thickness of HfO2 can increase the total resistance of HfO2/TiOx. Can you explain the relationship with the TiOx thickness and the diameter of conductive filament in more detail?

2. As shown in Figure 2, the reset process of the device shows gradual switching, whereas the reset process in Figure 2i displays the abrupt switching based on Joule heating. Why are there two different reset process happening? The authors should explain the mechanism of the reset process in more detail for the non-expert reader.

3. Through annealing process, the nanocrystal of HfO2 and TiOx are formed. But, the high temperature by annealing process is applied to the entire film region not the local region as shown in Figure 2d. So, the schematic of nanocrystal of HfO2 and TiOx should be modified.

4. (Page 5) What is a oxygen affinity electrode?

5.  (Page 2) a typing error : The author mentioned that “the nanocrystallin TiOx and HfO2e dots~”. Are “nanocrystalline” and “HfO2” correct instead of “nanocrystallin” and “HfO2e”?

6. (Page 4, Figure caption) a typing error : “H12T3 memristor before andn after annealing;~”. Is “and” correct instead of “andn”?

7. (Page 4) The authors mentioned nc-HfOx/TiOx device. But, the authors must write down the full name of nc.

8.  (Page 5) a typing error : “theorigin of the mobile electrons”. Is “theorigin” correct instead of “the origin”?

9. (Page 5) a typing error : “the nanopathway were accurately neutralize by~”. Is “neutralize” correct instead of “neutralized”?

Reviewer 3 Report

The paper entitled as, “Artificial HfO2/TiOx synapses with controllable memory window and high uniformity for brain-inspired computing”, discuss the controllable memory window of HfO2/TiOx memristive device can be obtained by tuning the thickness ratio of sublayer. The work seems to be good and can be considered for publication after incorporating following recommendations and answering a few related questions.

My decision at this stage is Minor Revision.

1-     The introduction section lacks intensive literature review. This article should direct its readers to those related works on memristive devices that can benefit them in their research work. Authors are recommended to add the following recently published papers on biomaterial based memristive devices in general and TiO2 based memristors in particular.

a.      Rehman, M.M., ur Rehman, H.M.M., Kim, W.Y., Sherazi, S.S.H., Rao, M.W., Khan, M. and Muhammad, Z., 2021. Biomaterial-based nonvolatile resistive memory devices toward ecofriendliness and biocompatibility. ACS Applied Electronic Materials, 3(7), pp.2832-2861.

b.     Jaafar, A.H., Al Chawa, M.M., Cheng, F., Kelly, S.M., Picos, R., Tetzlaff, R. and Kemp, N.T., 2021. Polymer/TiO2 Nanorod Nanocomposite Optical Memristor Device. The Journal of Physical Chemistry C, 125(27), pp.14965-14973.

c.      Ismail, M., Chand, U., Mahata, C., Nebhen, J. and Kim, S., 2022. Demonstration of synaptic and resistive switching characteristics in W/TiO2/HfO2/TaN memristor crossbar array for bioinspired neuromorphic computing. Journal of Materials Science & Technology, 96, pp.94-102.

d.      El Mesoudy, A., Lamri, G., Dawant, R., Arias-Zapata, J., Gliech, P., Beilliard, Y., Ecoffey, S., Ruediger, A., Alibart, F. and Drouin, D., 2022. Fully CMOS-compatible passive TiO2-based memristor crossbars for in-memory computing. Microelectronic Engineering, 255, p.111706.

e.      Khan, M., Mutee Ur Rehman, H.M., Tehreem, R., Saqib, M., Rehman, M.M. and Kim, W.Y., 2022. All-Printed Flexible Memristor with Metal–Non-Metal-Doped TiO2 Nanoparticle Thin Films. Nanomaterials, 12(13), p.2289.

2-     Grammatical mistakes should be reviewed closely throughout the manuscript before final acceptance.

3-     The conductive channels shown in TEM image of Figure 1(b) is not clear. How can the authors be sure about it that this is the conductive channel and not something else?

4-     There are several reports already available on TiO2 and HfO2 based memristors and RRAM devices. What are the standout features of this reported device in comparison to those previously reported devices?

5-     How have the authors calculated slope values of double logarithmic graphs shown in figure 3 (a-b)?

What was the yield of their memristive device fabrication?

Author Response

Please see the attatchment.

Round 2

Reviewer 1 Report

The authors have improved some parts of the manuscript in their revision.  It is adequate to be published.